# Middle Segment-Preserving Pancreatectomy to Avoid Pancreatic Insufficiency: Individual Patient Data Analysis of All Published Cases from 2003–2021

**DOI:** 10.3390/jcm12052013

**Published:** 2023-03-03

**Authors:** Thomas M. Pausch, Xinchun Liu, Josefine Dincher, Pietro Contin, Jiaqu Cui, Jishu Wei, Ulrike Heger, Matthias Lang, Masayuki Tanaka, Stephen Heap, Jörg Kaiser, Rosa Klotz, Pascal Probst, Yi Miao, Thilo Hackert

**Affiliations:** 1Department of General, Visceral and Transplantation Surgery, Heidelberg University Hospital, 69120 Heidelberg, Germany; 2Pancreas Center, The First Affiliated Hospital of Nanjing Medical University, Nanjing 210029, China; 3Department of Gastrointestinal Surgery, Affiliated Hangzhou First People’s Hospital, Zhejiang University School of Medicine, Hangzhou 310006, China; 4Department of Surgery, Keio University School of Medicine, Tokyo 160-8582, Japan; 5Study Center of the German Society of Surgery, University of Heidelberg, 69120 Heidelberg, Germany; 6Department of Surgery, Cantonal Hospital Thurgau, 8501 Frauenfeld, Switzerland

**Keywords:** pancreas/surgery, pancreatectomy/methods, pancreatectomy/adverse effects, retrospective studies

## Abstract

Middle segment-preserving pancreatectomy (MPP) can treat multilocular diseases in the pancreatic head and tail while avoiding impairments caused by total pancreatectomy (TP). We conducted a systematic literature review of MPP cases and collected individual patient data (IPD). MPP patients (N = 29) were analyzed and compared to a group of TP patients (N = 14) in terms of clinical baseline characteristics, intraoperative course, and postoperative outcomes. We also conducted a limited survival analysis following MPP. Pancreatic functionality was better preserved following MPP than TP, as new-onset diabetes and exocrine insufficiency each occurred in 29% of MPP patients compared to near-ubiquitous prevalence among TP patients. Nevertheless, POPF Grade B occurred in 54% of MPP patients, a complication avoidable with TP. Longer pancreatic remnants were a prognostic indicator for shorter and less eventful hospital stays with fewer complications, whereas complications of endocrine functionality were associated with older patients. Long-term survival prospects after MPP appeared strong (median up to 110 months), but survival was lower in cases with recurring malignancies and metastases (median < 40 months). This study demonstrates MPP is a feasible treatment alternative to TP for selected cases because it can avoid pancreoprivic impairments, but at the risk of perioperative morbidity.

## 1. Introduction

Total pancreatectomy (TP) can be the standard surgical treatment for a range of pathologies, including cancer, chronic pancreatitis, and intraductal papillary mucinous neoplasia (IPMN) [1,2,3,4]. TP is also conducted to avoid high-risk pancreatoenteric anastomosis or as a rescue operation for partial pancreatectomy [5,6]. Compared to partial pancreatectomy, TP can benefit from the absence of postoperative pancreatic fistula (POPF) and subsequent secondary complications that can otherwise complicate the short-term postoperative course (e.g., erosional post pancreatectomy hemorrhage) [7]. Nevertheless, TP also has serious postoperative mortality that increases with the extent of associated vascular/multi-visceral resection [8,9]. Furthermore, over the long-term, TP can result in physical impairment and reduced quality of life (QOL) due to deficiencies of the endocrine and exocrine systems (e.g., diabetes mellitus, DM) [10,11,12]. Consequently, there is an unmet need and responsibility to consider saving unaffected pancreatic parenchyma to avoid the negative functional consequences that can occur after removing the whole pancreas.

The time is ripe for such amendments, as the outcomes of pancreatic surgery are improving due to technological advancement (e.g., surgical instruments and techniques) and healthcare optimization (e.g., hospital centralization, patient-selection, perioperative and intensive care). New techniques include parenchyma-sparing resections of the pancreatic head [13] and body [14] as well as minimally invasive local excisions of small benign/premalignant lesions [15,16]. Such procedures can serve as components of middle-preserving pancreatectomy (MPP), or middle segment-preserving pancreatectomy (MSPP; hereafter MPP). This relatively new operation for the resection of multilocular diseases spares unaffected parenchyma along the pancreatic body (Figure 1). An initial publication reported two sequential operations for recurrent pancreatic carcinoma in 1999 [17] and the first one-stage procedure was reported in 2003 [18]. However, this procedure is still rare and lacks evidence as a viable alternative to TP.

Our study aimed to collect and summarize the current literature on MPP in terms of its perioperative parameters and postoperative outcomes. By doing so, we intended to (i) provide a thorough description of all MPP operations published so far, and (ii) generate the highest quality evidence possible on its surgical and clinical outcomes, given the limitations for analyzing such a rare procedure.

## 2. Materials and Methods

To achieve these aims we conducted a literature search for all MPP surgical cases published to date. As there have been no controlled trials for MPP, we requested complete individual patient data (IPD) from the original authors to generate a sample conducive to statistical analysis [19]. We included a comparative benchmark of TP patients curated from a database at a high-volume center for pancreatic surgery in order to enable a meaningful interpretation of the outcomes.

The nature of case-study data is not amenable to standard meta-analysis or mixed-model techniques because reports mostly cover single individuals and contain neither repeated measurements within patients nor multiple treatment arms. Therefore, we effectively ignored clustering and treated every patient as an independent case sampled from a random population. The IPD approach has advantages over analyzing aggregate data by standardizing the methodology across studies while avoiding pitfalls that stem from selection or publication biases. Furthermore, the dataset can involve more outcome measurements than those reported in source publications. Thus, analyses may be more reliable than a meta-analysis of aggregate data [19]. The study protocol was registered with Prospero (CRD42018112324) and performed per PRISMA and MOOSE guidelines [20,21,22].

### 2.1. Search Strategy and Eligibility Criteria

The Web of Science and MEDLINE databases were systematically searched using OVID (Heidelberg University), PubMed, and Clarivate between 5 April 2018 and 31 January 2022, following the guidelines of Kalkum et al. [23]. The search terms used were “middle segment preserving pancreatectomy” or “middle preserving pancreatectomy” in the title or abstract, without any other limitations. Further articles were manually sourced from the reference lists of retrieved articles if they mentioned the above terms.

The included studies were full-text reports of MPP surgeries and survival outcomes. The exclusion criteria were: (i) duplicate studies, (ii) reviews without original data, (iii) animal studies, (iv) absence of individual patient data, and (v) MPP performed in a two-stage procedure. Two-stage procedures were excluded because they represent sequential partial pancreatic resections (often in cases of recurrence). These contrast with the focal procedure, which attempts to remove synchronous multilocular diseases in one operation. Investigators TMP, JD, and XL independently screened all retrieved articles according to the eligibility criteria. Disagreements between reviewers were resolved by discussion until a consensus was found. The original authors were contacted for missing patient information, and articles were excluded from the analysis if none were provided. 

A comparative benchmark group of TP patients was selected from a homogenous single-center cohort. Thus, patients in the sample underwent relatively standardized procedures and the error attributable to variation in techniques across centers was limited. Patients for the TP group were extracted from Heidelberg University Hospital’s patient database of 244 patients who underwent TP between February 2002 and November 2020 due to benign multilocular pancreatic pathologies. Only patients with benign pathologies were selected in order to conservatively provide a benchmark with less oncological risk, thereby sharpening the contrast with MPP patients. Patients were selected from the database by retrospective assessment for the possibility of MPP by investigators TMP and JD based on careful examination of preoperative radiological images, intraoperative surgical evaluations, and postoperative histopathology results. This curation provided some approximation of patient matching and ensured that the samples were comparable. There were no criteria for minimum follow-up times for either the MPP or TP samples.

### 2.2. Quality Control and Risk of Bias

IPD were manually checked for consistency and completeness by investigators TMP, JD, and XL, with disagreements resolved by consensus. The methodological quality of each included study was assessed by investigators TMP and PP using a risk of bias evaluation tool for case reports and series, adapted from the ‘Critical Appraisal Skills Program’ and the ‘Newcastle Ottawa Scale’ [24]. Investigators TMP and PP also considered the overall dataset in light of the 7 domains of the ROBINS-I tool [25].

### 2.3. Extracted Data and Definitions

The full extracted data terms are available in Appendix A. They can be broadly classified as (i) study-level data, (ii) patient baseline characteristics, (iii) surgical procedures performed, (iv) intra-operative outcomes, (v) postoperative course and complications, and (vi) survival information. Data were collected by investigators TMP, JD, and XL, with disagreements resolved by consensus. If the information was not provided in the primary source, classifications were made via correspondence with the original authors or the data was left missing if no information could be retrieved. 

Pathologies indicated for pancreatic resections can be highly diverse. Therefore, we arbitrarily classified them into 4 groups. The first included ‘multifocal primary pancreatic neoplasms’, such as pancreatic ductal adenocarcinoma and IPMN (neoplasia group). Importantly, this general definition of neoplasia covered any pathology associated with abnormal tissue growth, benign or malignant. The second group included ‘multifocal pancreatic lesions that have metastasized from other organs’, including renal cell carcinoma and colorectal cancer (metastatic group). The third group, ‘synchronous heterogeneous pancreatic diseases’, included those in which the pancreatic head and tail had different pathologies (synchronous group). The final group was ‘non-neoplastic pancreatic diseases’, primarily chronic pancreatitis.

### 2.4. Outcomes and Analysis

#### 2.4.1. Patient Baseline Characteristics

Patient characteristics were summarized and reported descriptively, but statistical tests were used to determine if there were any potential confounding differences between MPP and TP samples (continuous variables: *t*-test; categorical variables: Chi-squared or Fisher’s exact tests).

#### 2.4.2. Surgical Resection and Intraoperative Outcomes

The intraoperative outcomes for MPP were summarized descriptively based on the specific procedures used during operations. Statistical models were untenable in this regard because the number of replicates under each procedure was not sufficient. However, we compared intraoperative outcomes between non-expanded pancreatic resections and multi-visceral resections with *t*-tests. We compared outcomes with the TP group by pooling all MPP operations together and using Type III ANOVAs, incorporating age, American Society of Anesthesiologists classification (ASA°) [26,27,28], and sex as patient-level covariates. These analyses were repeated using a reduced dataset that excluded multi-visceral resections from the MPP sample, since all TP operations were conducted around the pancreas and the extent of resection surgery can have significant associations with intraoperative outcomes [8,9]. Resections of the spleen, duodenum, and gallbladder were excluded from the definition of multi-visceral resection as these are part of standard pancreatic resection.

#### 2.4.3. Postoperative Course

The postoperative course for MPP patients was summarized descriptively and formally analyzed using logistic regression models for the effects of patient characteristics, surgical outcomes, and pathological dignity on binary event outcomes. A negative binomial generalized linear model (GLM) was used to analyze the length of stay because the Poisson model showed overdispersion. A full model containing all factors of interest could not be fitted for many of these models, so three reduced models (patient characteristics, surgical outcomes, pathological dignity) were fitted and compared to a null model with no factors using the corrected Akaike Information Criterion (AIC). Models were interpreted if they (i) provided equivalent or better explanatory power than the null model, (ii) satisfied model assumptions, and (iii) contained (borderline) significant effects from any component parameter based on a Wald test or the confidence interval. 

Postoperative outcomes for patients in the MPP and TP groups were compared using logistic regression models (negative binomial GLM for the length of stay), with the surgery group as the treatment of interest and age, ASA°, and sex as patient-level covariates. The main effect of surgery type and the confidence interval of its odd ratios were used to infer any differences in outcome between surgery groups. Models could not be fitted in comparisons where all patients in a group had the same outcome, so some comparisons were made descriptively.

#### 2.4.4. Long-Term Follow-Up and Survival

Robust models of survival analysis were untenable because of the wide variation in follow-up time, small sample size, and relatively few deaths observed. Therefore, our survival analysis for patients receiving MPP was based on summary descriptions of the patients who had died and univariate comparisons with those who survived. We used a reduced dataset to provide some control over the variation in follow-up times by excluding patients without an observed death who had less than 10 years of follow-up. We provided tentative survival estimates for MPP patients using Kaplan-Meier models.

## 3. Results

### 3.1. Literature Review and Risk of Bias

All of the included studies were case reports or small case series (Table 1). Of the 31 articles identified through the initial search, 17 articles with a total of 28 MPP patients were included in the final analysis (Figure 2). Of these articles, 11 were assessed to have a moderate risk of bias (Table A1) because the authors did not report the selection of patients included in the article (i.e., the study may have omitted other patients undergoing MPP at the same center). Other domains for risk of bias at the study level were assessed as negligible. We completed our analytical sampling with patients from our records. Here, we added 1 additional patient to the MPP group and formed a comparator group of 14 TP patients that had been evaluated as potential MPP candidates, bringing the IPD to 43 patients (29 MPP and 14 TP).

We determined that risk of bias may exist at the sample level for 2/7 domains of ROBINS-I [25]. Firstly, our samples may have carried serious risks of bias due to confounding because the patient characteristics (e.g., age, comorbidity, suspected histology) contributed to both the chosen intervention and surgical outcome. Furthermore, our TP sample represented non-independent homogeneous operations from a single center, whereas the MPP sample represented diverse and independent operations from multiple centers. Secondly, we also assessed a moderate risk of bias due to deviations from the intended interventions; the MPP sample contained different co-interventions and intra-operative technical variations, whereas procedures were more standardized in the TP group. We could partly account for these risks by including patient-level covariates in the analysis, but they were otherwise inherent limitations to a retrospective case study analysis with a curated comparator group.

### 3.2. Pre-Operative Baseline

IPD for baseline variables are provided in Appendix A and a summary for the MPP and TP groups is provided in Table A2. A summary of the underlying pathologies in the pancreatic head and tail is provided in Table A2. Across all patients, the mean age at surgery was 61 years, with 24 females (56%) and 19 males (44%). ASA° was ≤2 for 27 patients (63%) and >2 for 16 patients (37%). Pre-existing diabetes mellitus was present in 8/43 patients (19%). There were no significant differences in the mean age or distributions of sex, ASA°, or pre-existing diabetes between groups (Table A2). The most common surgical indication was for pancreatic neoplasia, in both MPP (12/29; 41%) and TP (8/14; 57%) patients. Multiple synchronous pancreatic pathologies were also common among both MPP (10/29; 34%) and TP (3/14; 21%) patients. The remaining 3 (2%) TP patients were indicated for non-neoplastic pathologies (mainly chronic pancreatitis), compared to 1 (3%) MPP patient. There were 6 (21%) MPP patients indicated for intra-pancreatic metastatic pathologies, but none from the TP group. This is because only patients with benign multilocular pancreatic pathologies were selected for the TP group and hence no patients were indicated for metastatic pathologies. The result was a borderline significant difference in the distribution of surgical indications between groups (Table A2). Relatedly, 19/29 (66%) MPP patients had a malignant pathology, whereas all 14 TP patients had a benign pathology (Fisher’s test: *p* < 0.001). 

### 3.3. Surgical Resections and Intraoperative Outcomes

Individual patient data for surgical procedures and intraoperative outcomes are provided in Appendix A. The procedures are illustrated in Figure 1 and the intraoperative outcomes are summarized in Table 2. The intraoperative outcomes are compared between MPP and TP groups in Figure 3 and Table A4.

#### 3.3.1. MPP Surgical Procedures

The most common proximal MPP operation was pylorus-preserving pancreaticoduodenectomy (PPPD; Traverso-Longmire procedure; 12/29; 41%). Other common procedures included pancreaticoduodenectomy (PD; Kausch-Whipple procedure; 7/29; 24%), and subtotal stomach-preserving pancreaticoduodenectomy (SSSPD; 5/29; 17%), 1 of the latter being laparoscopic. Parenchyma-sparing resections at the head occurred in 5/29 (17%) cases, including 2 inferior pancreatic head resections, 2 duodenum-preserving pancreatic head resections (including 1 Beger procedure), and 1 uncinate process resection. Head resections were coupled with spleen-preserving distal pancreatectomy or spleen-resecting procedures, which were used independently of the proximal procedure (Fisher’s test: *p* = 0.260). Most operations (24/26; 92%) used pancreaticojejunostomy for anastomosis, whereas 1 operation used reversed pancreaticogastrostomy and 1 operation did not use anastomosis (after uncinate process resection). Transection was typically achieved with a scalpel and sutures (15/21; 71%), with a stapler used in 6/21 patients (29%).

Multi-visceral resections in addition to the pancreatic resection (6/29; 21%) included: superior mesenteric vein resection and reconstruction, right lobectomy of the liver, left hepatectomy due to single liver metastasis, and right hemicolectomy. There were also two metachronous and synchronous resections for extrapancreatic lesions, including a metastatic dermatofibrosarcoma protuberans (DFSP) in the right lung and a metastatic pheochromocytoma at a previous surgical site of left adrenalectomy.

#### 3.3.2. MPP Intraoperative Outcomes

The median operation time for MPP surgeries was 440 min (range: 250–670 min). Multi-visceral operations tended to run longer than operations restricted to the pancreas, although not significantly (t_6.6_ = 1.54, *p* = 0.169). Operations were significantly longer in older patients (Figure 3a, Table A4). There were 2 patients with statistically outlying amounts of blood loss at 5055 and 5500 mL. Median blood loss was 800 mL (range: 150–5500 mL), with no significant difference in blood lost between multi-visceral and pancreatic resections (t_5.8_ = 0.88, *p* = 0.414). Significantly more blood was lost in patients with ASA° > 2 (Figure 3b, Table A4). The remnant pancreas had a median length of 5.1 cm (range: 2.0–9.0 cm, N = 27) and a median of 33.8% of the original volume (range: 15.0–56.9%, N = 18). Patient variables had no significant association with remnant length (Table A4). We detected significantly longer pancreatic remnants when the resection was restricted to the pancreas and was not multi-visceral (t_15.0_ = 2.20, *p* = 0.044), but this pattern did not hold for remnant volume (t_6.0_ = 0.15, *p* = 0.885).

#### 3.3.3. Comparison of Intraoperative Outcomes with TP Patients

The mean log operation time was significantly longer in the MPP group than in the TP group (Figure 3a, Table A4), but blood loss did not significantly differ between surgery types (Figure 3b, Table A4). Remnant lengths ranged from 2.0 to 9.0 cm (Q1–Q3: 4.75–7.00 cm) and estimated unaffected parenchyma lengths ranged from 3.0 to 9.0 cm (Q1–Q3: 3.0–5.0 cm). There was no significant difference between the mean length of the remnant in MPP patients and the estimated unaffected parenchyma in TP patients (Figure 3c, Table A4).

### 3.4. Postoperative Course and Complications

Individual patient data for the postoperative course are provided in Appendix A. The observed frequencies for short- and intermediate-term complications in the MPP and TP groups are provided in Figure 4. Figure 5 summarizes the prognostic indicators for complications in the MPP group. The results of multi-model comparisons and the estimates of selected models are provided in Table A5, Table A6, Table A7 and Table A8. Comparisons between MPP and TP groups are available in Table A9.

#### 3.4.1. MPP Postoperative Outcomes

The median postoperative length of stay was 30 days (min, Q1, Q3, max = 5, 21, 50, 139 days, respectively). Hospital stays were significantly longer after prolonged operations and operations with greater amounts of blood loss (Figure 5a). Although 69% of MPP patients experienced a postoperative event and 76% experienced morbidity (Figure 4), most of the complications were of minor to intermediate severity. POPF occurred in more than half of the MPP patients, but all cases were POPF Grade B, and all patients recovered completely after interventional drainage placement. There were no subsequent life-threatening complications, no cases of organ failure, and no rescue operations were needed (i.e., no Grade C cases). Delayed gastric emptying occurred in 14% of patients, but all cases were successfully managed with conservative treatment. Other complications occurred in 31% of patients, including symptomatic pseudocyst, respiratory failure, early post-pancreatectomy hemorrhage (Type B PPH), splenic hematoma, peritoneal bleeding, liver abscess, cholangitis, pleural effusion, and superficial wound-healing disorder. Three patients (10%) had recorded readmissions, all of whom were treated and cured for other complications (symptomatic pseudocyst, liver abscess, and cholangitis). 

Multi-model comparisons indicated that the surgical outcome provided notable explanatory power for an uneventful postoperative course, morbidity, POPF, and readmission (Table A5). Specifically, the odds of an uneventful course (borderline) improved with the length of the pancreatic remnant, and longer remnants were also (borderline) associated with lower rates of morbidity and POPF (Figure 5b–d).

#### 3.4.2. MPP Postoperative Pancreatic Function

Endocrine pancreatic insufficiency was reported in 44% of patients, including also mild cases of impaired glucose tolerance and fasting glucose (Figure 4). New-onset diabetes mellitus occurred in 7 (29%) patients who had no pre-existing condition, with 5/7 (71%) being insulin-dependent and 2/7 (29%) being non-insulin-dependent. Exocrine insufficiency was recorded in 8/27 (30%) patients and presented as steatorrhea after fatty food (7/8; 88%) and decreased fecal elastase (<50 μg/g; 1/8; 13%).

Multi-model analysis indicated that patient variables were informative for explaining the development of endocrine insufficiency and new-onset DM, but not exocrine insufficiency (Table A5 and Table A6). In particular, the odds of presenting with endocrine insufficiency or DM following MPP surgery increased with age (Figure 5e,f). The model for endocrine insufficiency also detected a borderline effect of ASA° because 10/12 (83%) patients with endocrine insufficiency had ASA° ≤ 2. In contrast, multi-model comparisons suggested that pathological dignity was informative for explaining exocrine insufficiency (Table A5). A post-hoc analysis revealed that these results were heavily influenced by patients with chronic pancreatitis (CP). Specifically, non-CP patients experienced only 6% (0–81%) the rate of exocrine insufficiency as CP patients (Fisher’s test: *p* = 0.015, Figure 5g). Sensitivity analyses that removed CP patients suggested that the surgical outcome model was informative, but no component factors had clear prognostic associations with exocrine insufficiency (Table A5 and Table A7). Finally, all patients with exocrine insufficiency received operations that resected or reduced the pancreatic head (6 PPPD; 1 PD; 1 DPPHR). 

#### 3.4.3. Comparison of Postoperative Course with TP Patients

MPP patients had significantly longer postoperative stays than TP patients, but there were no significant differences in the likelihood of adverse events or morbidity between groups (Table A9). Although the overall incidence rates were similar, the rates for specific complications varied between groups (Figure 4, Table A9). On the one hand, readmissions, POPF, and DGE only occurred in the MPP group and were completely absent in the TP group. On the other hand, less than half of the MPP patients developed endocrine or exocrine pancreatic insufficiency, whereas both of these conditions were prevalent in the TP group. Moreover, few MPP patients suffered from manifest diabetes mellitus or insulin dependency. Of the data available, 1/7 (14%) MPP patients experienced a hypoglycemic event compared to 10/10 TP patients (Fisher’s test: OR = 0.00–0.27, *p* < 0.001). MPP patients also displayed lower mean HbA1c (t_18.06_ = 2.13, *p* = 0.047) and fasting blood glucose (t_8.04_ = 13.07, *p* < 0.001) than TP patients. MPP patients were significantly less likely to experience exocrine insufficiency than TP patients (Table A9), but there was no significant difference in their presentation. The rate of steatorrhea did not differ (Fisher’s test: OR = 0.10–2.72, *p* = 0.480) between MPP (7/17; 41%) and TP (8/14; 57%) patients. Average weight loss was 4.00 ± 5.68 kg for MPP patients (N = 8) and 7.85 ± 8.56 kg for TP patients (N = 14), but this difference was not significant (t_19.38_ = 1.26, *p* = 0.221). Finally, patients in the MPP group were significantly less likely to present with other complications than those in the TP group (Figure 4, Table A9).

### 3.5. MPP Long-Term Follow-Up and Survival Analysis

Median follow-up for the MPP group was 72 months (range: 5–228), in which 6 patients (21%) died (Table A10). Malignancy was assessed in all 6 patients, with a metastatic stage present at the time of operation in 3 cases (renal cell cancer, DFSP, and rectal cancer). Of those that died, 4/6 (67%) patients died from tumor recurrence or progression to systemic metastases. Of the remaining 2 patients, one died from cerebral infarction and the other from malignant lymphoma. Of the 5/29 MPP patients with an observed recurrence of malignancy, 4/5 (80%) patients died during follow-up. 

The majority of malignant cases survived over the follow-up period (13/19; 68%), although follow-up times ranged from 5 to 185 months. Kaplan-Meier estimates suggested that median survival among all MPP patients with malignant pathologies was 110 months (95% CI: 58—incalculable). One-year survival for malignant cases undergoing MPP was 94% and three-year survival was 81%.

To control for variation in follow-up times, we used a reduced dataset including only patients with survival follow-up extending to at least 10 years (N = 12). Univariate comparisons between those that died or survived indicated that survival may be associated with the recurrence of malignancy after surgery and pancreatic remnant length (Table 3). Specifically, those who experienced a recurrence were borderline more likely to die within 10 years, while patients with larger remnants were significantly more likely to survive than die within 10 years. There was no significant difference in survival between malignant and non-malignant cases. Within the reduced dataset, median survival for those with malignant pathologies at the time of surgery was estimated at 58 months, whereas it was estimated at 39 months for patients with recurrent malignancies. Patients with remnants smaller than the overall MPP median (<5.15 cm) were estimated to have a median survival of 36 months. Median survival across all 12 patients in the reduced dataset was estimated at 110 months. One-year survival within the reduced dataset was 92% (CI: 77–100) and three-year survival was 75% (CI: 54–100).

## 4. Discussion

MPP is a novel and rare surgical technique that demands investigation. By compiling all published reports on the procedure, our dataset provides an early opportunity to study its potential risks and benefits. Although MPP procedures take longer on average than TP, it appears to be effective at sparing unaffected parenchyma that would otherwise be lost. This can have significant benefits for the postoperative course and long-term outcomes. Specifically, MPP is associated with severely reduced risks of functional endocrine and exocrine complications, including new-onset DM. That being said, MPP does not eliminate the risks of POPF or DGE as completely as TP. The long-term outcomes after MPP surgery appear promising, especially for patients with benign pathologies and large spared remnants. Survival for patients with malignant pathologies was conservatively estimated at 58 months, despite many of these patients being metastatic at the time of surgery. Overall, this systematic review and analysis suggest that MPP is a feasible alternative to TP in light of ongoing developments in pancreatic surgery. Furthermore, this early analysis provides reliable evidence for more dedicated clinical research efforts. However, we must stress that potential cases for MPP are still expected to be rare since our evaluation of 244 patients receiving TP for multilocular disease over 18 years revealed only 14 (6%) patients suitable for MPP. This indicates that there is most often no alternative to TP. 

TP is the current standard of care for multilocular pancreatic pathologies (even if lesions spare the pancreatic body) because of its capacity to avoid POPF and other secondary complications in the short term. However, it comes with physical impairment, high mortality rates, and reduced QOL in the long term [10,11,12]. Major concerns include diabetes, with a systematic review finding that approximately 80% of TP patients develop hypoglycemic episodes (with 40% experiencing severe hypoglycemia), resulting in 0–8% mortality and 25–45% morbidity [10]. Pancreoprivic diabetes can complicate recovery and predispose patients to readmission, leading to treatment costs that can be triple those for patients with non-pancreatogenic diabetes [12]. Fortunately, the management of endocrine insufficiency after TP has improved over the past decades, leading to diabetes-specific outcomes that seem equivalent to other types of insulin-dependent diabetes [51]. Additionally, QOL impairment by exocrine insufficiency might be improved by modern pancreatic enzyme preparations [52]. However, endocrine and exocrine insufficiencies still seem to heavily impair QOL after TP [53,54,55]. The life-long burdens related to TP were illustrated in a recent study reporting that the psychosocial impact of diabetes, the need for insulin therapy, and the severity of exocrine insufficiency were all significantly greater after TP than after a Kausch-Whipple procedure [5]. Therefore, there are severe detriments to the complete removal of the parenchyma and a consequent need to critically evaluate the application of TP as a treatment or rescue operation in selected cases. Indeed, it has been recommended to use TP in non-oncological cases only when avoiding POPF and its related sequelae can overcome the drawbacks of life-long diabetes and exocrine insufficiency [5].

Functional complications of the endocrine and exocrine pancreatic systems are a ubiquitous concern for TP patients [10,11,12], and MPP represents a method with the potential to retain pancreatic functionality. The pre-operative rates of DM were equivalent between MPP and TP patients, implying a similar baseline quality of the pancreatic parenchyma between groups. Therefore, the subsequent difference in new-onset DM after surgery suggests that patients may be saved from postoperative diabetes by receiving MPP. Additionally, pancreoprivic diabetes may be more serious after TP, as potentially life-threatening hypoglycemic events were common among TP patients but rare among MPP patients. However, older patients were still more likely to experience functional complications than younger patients after MPP in terms of endocrine insufficiencies and new-onset diabetes, possibly due to the natural degradation of organ quality with age. Additionally, the risk of exocrine insufficiency may be greater for patients with chronic pancreatitis due to disease-related organ damage. The risk of exocrine insufficiency may also increase with the extent of resection of the pancreatic head. In total, however, the majority of patients were free of functional endocrine and exocrine complications after MPP surgery, whereas they were guaranteed after TP surgery.

In contrast to TP, MPP carries the risks of typical complications of partial pancreatectomy, such as POPF and DGE. In this respect, remnant length was highlighted as an important prognostic factor following MPP. Specifically, longer pancreatic remnants were associated with shorter and less eventful hospital stays alongside decreased risks of morbidity and POPF. However, it is unclear whether saving more parenchyma necessarily translates into better clinical outcomes, as it may also be easier to preserve longer remnants in healthier patients with good baseline prognoses. Remarkably, our evidence emphasizes that larger pancreatic remnants are not associated with greater risk of developing POPF or other complications compared to smaller remnants.

Overall, 6 MPP patients died during follow-up. All of these patients had malignant pathologies, half of whom were already in a metastatic stage at the time of surgery. Indeed, death was significantly associated with recurrence of malignancy or metastatic progression. The length of the pancreatic remnant was also highlighted as a significant prognostic factor, as patients with longer remnants were significantly more likely to survive for 10 years than those with smaller remnants. However, this retrospective study cannot determine whether leaving longer remnants contributes to survival or if longer remnants can be saved in patients with better survival prospects. Overall, survival outcomes of MPP were promising, with a median survival of 110 months in a dataset covering 10 years of follow-up and three-quarters of patients suffering from malignant diseases. However, the number of reported MPP cases was still too small for thorough survival analysis.

Our analytical approach was limited by small sample sizes, but this was not a failure of study design. Indeed, the rarity of data and unmet need for synthesis was the issue being addressed by this research. Hence, our MPP sample included most, if not all, published and retrievable individual patient data available to date. This allowed us to establish the largest cohort of MPP patients possible for our analysis. In turn, our careful curation of a benchmark TP sample provided numerous control benefits at the cost of sample size and non-independence. Specifically, our sample benefited from limiting the error attributable to variation in techniques across centers. Furthermore, patients were carefully curated on the basis of whether they could have received MPP, thereby providing some approximation of patient matching and relevancy. Finally, the selection of TP patients with only benign cases produced a sample with less oncological risk for use as a healthier benchmark and to sharpen contrasts. Therefore, the small size and non-independence of the TP sample was not as serious a drawback as it appears since we were more interested in a reliable benchmark than in generalizing outcomes or effect measures to the general TP population. We highlight the limitations and biases inherent to our approach so that the results can be critically interpreted, but we also highlight that the analysis was designed to provide robust and reliable interpretations given the paucity of MPP data. Finally, compared to a meta-analysis of aggregate data, our IPD approach incorporated many outcome measurements whilst being more standardized and less impaired by selection or publication biases [19]. Nevertheless, we strongly recommend future research to improve upon our analytical methods as more MPP data becomes available. 

Our risk of bias analysis suggested that 65% of retrieved papers had a moderate risk of bias because they did not report on previous experience with MPP intervention. However, given the rarity of this operation and the value of reporting its outcomes, it is probably safe to assume that the reported cases reflected the entire experience of the responsible clinics. Even if this was not the case, the risk of bias existed only in the domain of patients selected to report, and we did not assess any risk owing to the quality of outcome measurements, alternative causalities, lack of follow-up, or fidelity in reporting.

## 5. Conclusions

This systematic review and IPD analysis suggest that MPP is a feasible alternative to TP. Perioperative and postoperative parameters were satisfactory, especially for younger patients and those with larger spared remnants. Furthermore, survival prognosis was satisfactory for those with benign disease and acceptable even for many with malignant diseases (providing there was no metastasis or recurrence). However, the number of reported cases was too low to make reliable survival estimates. In summation, despite some risk of short-term complications, MPP offers a resection technique that can spare pancreatic functionality and provide long-term benefits over TP. Therefore, it could be considered in rare and selected cases (e.g., young patients with large savable remnants, no pre-existing diabetes, and no organ damage from chronic pancreatitis). 

## Figures and Tables

**Figure 1 jcm-12-02013-f001:**
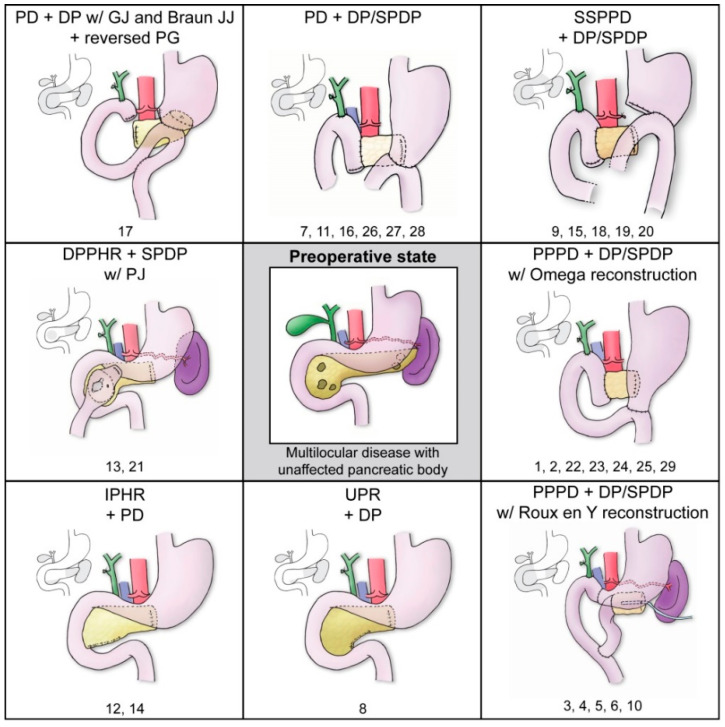
Surgical procedures used for middle-preserving pancreatectomy (MPP), showing final outcomes compared to the preoperative state in the middle. Smaller illustrations in grey indicate the resection targets. Structures displayed in the figures are: (i) pancreas (yellow) with multilocular disease (brown spots), (ii) gastroenteric tract (pink), (iii) gallbladder/ligated cystic bile duct and extrahepatic bile ducts (green), (iv) spleen (purple), (v) aorta with celiac trunk and splenic artery (red), (vi) portal vein (blue), and (vii) temporary gastric tube (light blue, attached to pancreatic stump in PPPD w/Roux en Y reconstruction). The numbers below the procedures are the associated patient IDs. DP: distal pancreatectomy with splenectomy. DPPHR: duodenum preserving pancreatic head resection. GJ: gastrojejunostomy. IPHR: inferior pancreatic head resection. JJ: jejunojejunostomy. PD: pancreaticoduodenectomy. PG: pancreaticogastrostomy. PJ: pancreaticojejunostomy. PPPD: pylorus-preserving pancreaticoduodenectomy. SPDP: spleen preserving distal pancreatectomy. SSPPD: subtotal stomach preserving pancreaticoduodenectomy. UPR: uncinate process resection.

**Figure 2 jcm-12-02013-f002:**
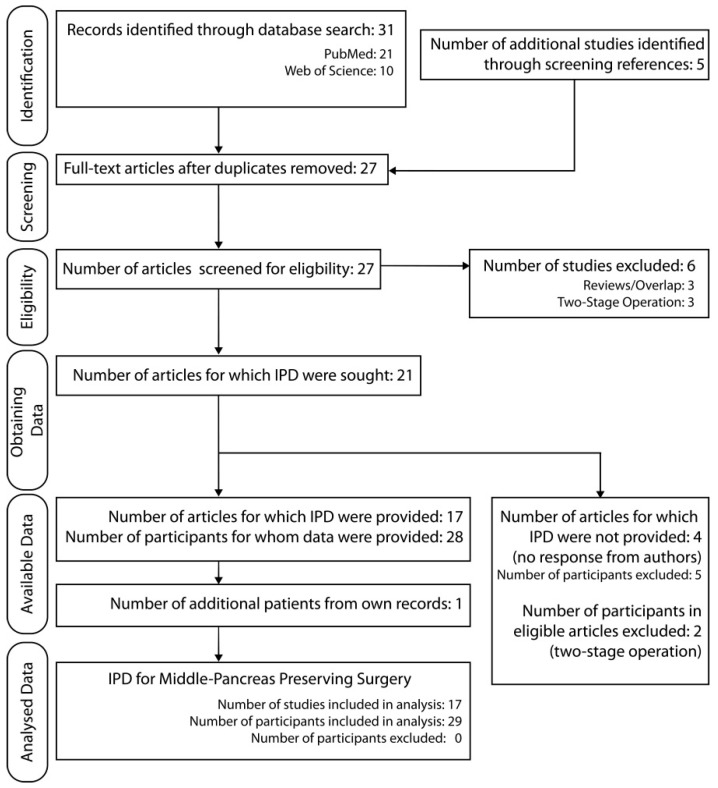
PRISMA-IPD flowchart for literature review and generation of MPP dataset. IPD: individual patient data.

**Figure 3 jcm-12-02013-f003:**
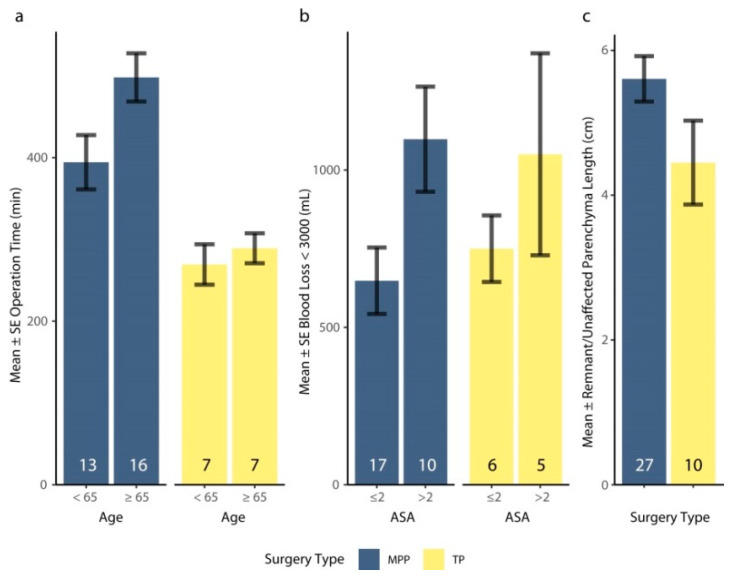
Mean ± SE (**a**) operation time (min) factored by age and surgery type, (**b**) blood loss (mL) factored by ASA classification and surgery type, and (**c**) length of the remnant (MPP) or estimated unaffected parenchyma (TP) by surgery type. The number of cases for each category is provided within the column. (**b**) Excludes 1 missing value and 2 patients with outlying blood loss. (**c**) Excludes 6 missing values for the respective response. ASA: American Society of Anesthesiologists. MPP: middle segment-preserving pancreatectomy. SE: standard error. TP: total pancreatectomy.

**Figure 4 jcm-12-02013-f004:**
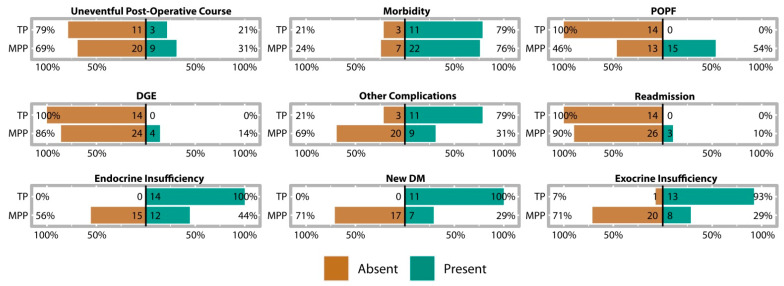
Within-group proportions for the presence or absence of postoperative complications during hospital stays. Frequency counts are provided within the bars. DGE: delayed gastric emptying. DM: diabetes mellitus. POPF: postoperative pancreatic fistula.

**Figure 5 jcm-12-02013-f005:**
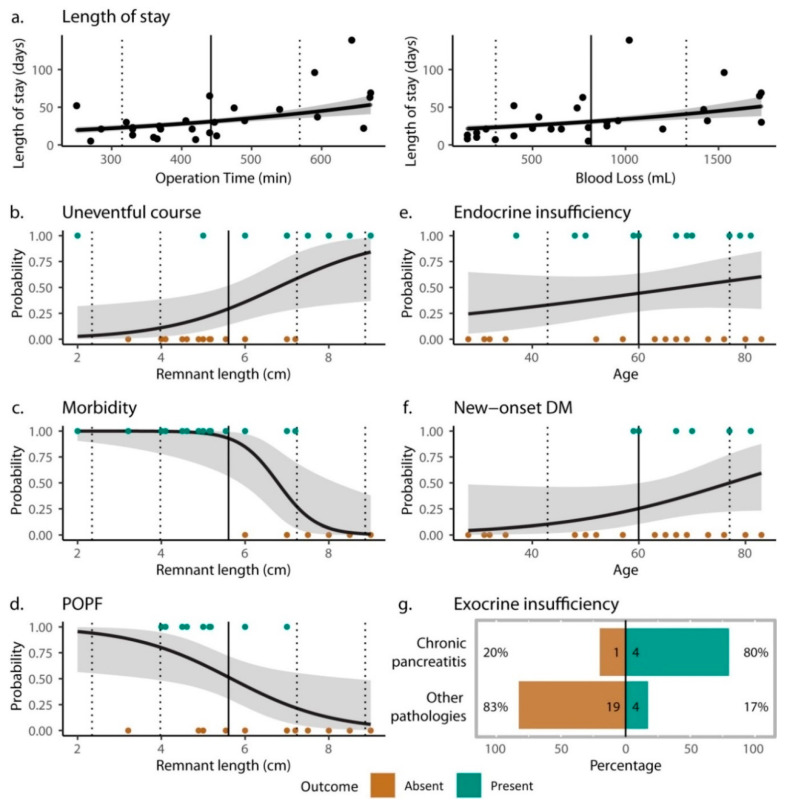
Summarized results for prognostic indicators of postoperative outcomes following MPP surgery: (**a**) length of stay by operation time and blood loss; (**b**) probability of uneventful postoperative course by remnant length; (**c**) probability of morbidity by remnant length; (**d**) probability of POPF by remnant length; (**e**) probability of endocrine insufficiency by age; (**f**) probability of new-onset diabetes mellitus by age; (**g**) frequency of exocrine insufficiency by chronic pancreatitis pathology. The bold curves represent the modelled probabilities for a ‘present’ response and shaded regions represent the 95% confidence interval; mean and standard deviation for the predictor variables are represented by solid and dotted lines, respectively; points represent observed outcomes; full results are available in Table A5, Table A6, Table A7 and Table A8. DM: diabetes mellitus. POPF: postoperative pancreatic fistula.

**Table 1 jcm-12-02013-t001:** Articles reporting MPP for which individual patient data (IPD) was sought, showing pathological indication and dignity of patients who received single-stage MPP and for whom IPD was obtained; also showing 1 MPP patient from the Heidelberg University Hospital database.

Ref.	Study ID	First Author	Year	Country	Indication	Pathological Dignity
[17]	NA	* Siassi	1999	Germany	NA	NA
[18]	1	Lloyd	2003	USA	Neoplasia	Benign
[29]	2	^†^ Miura	2007	Japan	Synchronous	Malignant
[30]	NA	^‡^ Chiang	2009	Taiwan	NA	NA
[31]	3	Partelli	2009	Italy	Neoplasia	Benign
					Neoplasia	Benign
					Neoplasia	Malignant
					Synchronous	Benign
					Synchronous	Benign
[32]	4	Kitasato	2010	Japan	Metastatic	Malignant
[33]	5	Ohzato	2010	Japan	Metastatic	Malignant
[34]	6	Sperti	2010	Italy	Synchronous	Benign
[35]	7	Chen	2011	China	Synchronous	Malignant
[36]	8	Horiguchi	2011	Japan	Neoplasia	Malignant
					Neoplasia	Malignant
					Neoplasia	Malignant
					Synchronous	Malignant
[37]	9	Noda	2011	Japan	Synchronous	Malignant
[38]	NA	^‡^ Otani	2011	Japan	NA	NA
[39]	NA	^‡^ Cheng	2013	China	NA	NA
[40]	10	Aryal	2014	Japan	Synchronous	Malignant
[41]	11	Nishi	2014	Japan	Neoplasia	Benign
[42]	NA	* Takeshi	2014	Japan	NA	NA
[43]	12	Tanemura	2014	Japan	Metastatic	Malignant
[44]	13	Usui	2014	Japan	Neoplasia	Benign
[45]	14	Lu	2016	China	Non-neoplastic	Benign
					Neoplasia	Malignant
					Metastatic	Malignant
					Metastatic	Malignant
					Metastatic	Malignant
[46]	NA	* Yamada	2017	Japan	NA	NA
[47]	NA	^‡^ Patyutko	2019	Russia	NA	NA
[48]	15	Addeo	2020	France	Neoplasia	Malignant
[49]	16	Nitta	2020	Japan	Neoplasia	Malignant
[50]	17	Iguchi	2021	Japan	Synchronous	Malignant
NA	NA	NA	2020	Germany	Synchronous	Benign

* Articles excluded from analysis because of two-stage operation. † 2 of 3 patients were excluded because of two-stage operation. ‡ Article excluded from analysis because individual patient data could not be obtained from authors. NA: not applicable.

**Table 2 jcm-12-02013-t002:** Summary of intraoperative variables for MPP patients by proximal operation and surgical extent; frequency counts separated by use of spleen-resecting or -preserving procedures at the distal end.

		DP	SPDP	Operation Time (min)	Blood Loss (mL)	Remnant Length (cm)	Remnant Volume (% Original)
		N (%)	N (%)	Mean ± SE	Mean ± SE	Mean ± SE	Mean ± SE
Proximal operation	Duodenum-preserving pancreatic head resection (Beger, Bern, Frey)	0 (0%)	2 (7%)	506.5 ± 136.5	1110.0 ± 90.0	4.8 ± 0.2	18.4
	Inferior pancreatic head resection	1 (3%)	1 (3%)	454.0 ± 86.0	1160.0 ± 260.0	NA	NA
	Pancreaticoduodenectomy (Kausch-Whipple procedure)	6 (21%)	1 (3%)	512.6 ± 34.6	1440.7 ± 632.6	5.3 ± 0.4	33.8 ± 4.0
	Pylorus-preserving pancreaticoduodenectomy (Traverso-Longmire procedure)	9 (31%)	3 (10%)	378.8 ± 38.0	981.6 ± 431.0	5.7 ± 0.6	28.6 ± 3.7
	Subtotal stomach-preserving pancreaticoduodenectomy	4 (14%)	1 (3%)	520.0 ± 48.0	885.8 ± 215.0	6.3 ± 0.7	33.4 ± 8.3
	Uncinate process resection	1 (3%)	0 (0%)	440.0	1720.0	5.0	40.0
Surgical extent	Pancreas only	17 (59%)	6 (21%)	430.3 ± 24.0	975.8 ± 215.4	5.9 ± 0.4	31.0 ± 3.2
	Multi-visceral	4 (14%)	2 (7%)	533.2 ± 62.1	1685.0 ± 777.3	4.7 ± 0.4	31.8 ± 4.7

DP: distal pancreatectomy with splenectomy. SE: standard error. SPDP: spleen preserving distal pancreatectomy.

**Table 3 jcm-12-02013-t003:** Frequency counts and means ± SE for MPP patients known to survive for 10 years (N = 6) and those who died within 10 years (N = 6).

	Survived	Died	Test	*p*-Value
Indication: Metastatic	1 (17%)	3 (50%)	Fisher’s	0.351
Indication: Neoplasms	4 (67%)	1 (17%)		
Indication: Synchronous	1 (17%)	2 (33%)		
Dignity: Benign	3 (50%)	0 (0%)	Fisher’s	0.182
Dignity: Malignant	3 (50%)	6 (100%)		
Malignancy recurrence: No	6 (100%)	2 (33%)	Fisher’s	0.061
Malignancy recurrence: Yes	0 (0%)	4 (67%)		
ASA° ≤ 2	5 (83%)	3 (50%)	Fisher’s	0.546
ASA° > 2	1 (17%)	3 (50%)		
Age	56.0 ± 7.8	71.3 ± 5.4	t_8.88_ = 1.61	0.141
Operation time (min)	462.2 ± 47.5	428.5 ± 65.7	t_9.10_ = 0.42	0.688
Blood loss (mL)	630.0 ± 212.8	1575.8 ± 720.4	t_5.87_ = 1.26	0.256
Remnant length (cm)	7.1 ± 0.6	4.9 ± 0.6	t_8.00_ = 2.46	0.040

ASA°: American Society of Anesthesiologists classification.

## Data Availability

Formatted tables of the data presented in this study are available in Appendix A; the full raw data are available from the corresponding authors upon reasonable request.

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
