# Peer review of "Middle Segment-Preserving Pancreatectomy to Avoid Pancreatic Insufficiency: Individual Patient Data Analysis of All Published Cases from 2003–2021"

_jcm, 2023, doi:10.3390/jcm12052013_

Round 1

Reviewer 1 Report

The authors stated that MPP is a feasible treatment for selected cases because it can preserve pancreas, but at the risk of perioperative morbidity comparing with TP.

Reviewer’s comment

1.     Patients number is too insufficient to conclude their insists.

2.     PD and other surgical method for pancreas head resection are different procedure and rate of postoperative complications. So, reviewer thinks that these are difficult to compare. 4/5 patients (80%) of MPP without PD had POPF.

3.     Reviewer did not understand why some reference with PDAC showed MPP in spite of no indication.

4.     Please show pancreas hardness or MPD size for MPP patients.

5.     What is TMP. Please clarify abbreviation.

Reviewer 2 Report

Dear Authors,

Thank you for the opportunity to review this interesting paper. It is concerned with a novel topic, well designed, with clearly described methodology and helpful figures and tables. The unavoidable weaknesses of the study are clearly pointed out and discussed. 

The manuscript is an analysis of individual patient data from the available literature on a rarely performed surgery, the middle segment-preserving pancreatectomy. The data was compared with total pancreatectomy outcomes from a single, large-volume center. The middle segment-preserving pancreatectomy may be an alternative to total pancreatectomy in selected patients. It reduces the rate of postoperative endo- and exocrine pancreatic insufficiency, however, has a higher risk of postoperative complications, such as postoperative pancreatic fistula. The authors applied appropriate methodology and described it clearly. The introduction is concise and informative. The text is supported with helpful, easy-to-understand figures, explaining the technical aspects of the demanding surgery. The manuscript is well-written, the figures and tables provide additional information and they are not redundant to the text. It is obvious that the authors have a thorough knowledge of the topic. The authors' conclusions are cautious and well supported by the study. The unavoidable weaknesses (e.g. small patient sample which is justified by the rarity of such operations) are acknowledged by the authors and properly discussed. The study is a valuable reminder that with the progress of surgical technique and preoperative care, middle segment-preserving pancreatic resection may be a better choice for selected patients with multilocular pancreatic disease.

Reviewer 3 Report

As the authors say, the main drawbacks of this manuscript are:

!. “TP sample represents non-independent homogeneous operations from a single center, whereas the MPP sample represents diverse and independent operations from multiple centers.”

2. “Secondly, we also assess a moderate risk of bias due to deviations from intended interventions because the MPP sample contains different co-interventions and intra-operative technical variations, whereas procedures were more standardized in the TP group.”

Another weakness is the small sample of TP patients. 

Section 2.1. Search strategy and eligibility criteria “Patients for the TP group were extracted from Heidelberg University Hospital’s pa-tient database of 244 patients who underwent TP between February 2002 and November 114 2020 due to benign multilocular pancreatic pathologies.“ Why only benign lesions?

3.2. Pre-operative baseline “The most common surgical indication was for pancreatic neoplasia, in both MPP (12/29; 41%) and 230 TP (8/14; 57%) patients.” And later in the same section “…all 14 TP pa-239 tients had a benign pathology” See previous comment. According to section 2.1 the TP group was selected among patients undergoing surgery for benign lesions. Please clarify.

Round 2

Reviewer 1 Report

The authors had corrected their manuscript as the reviewer commented.